# miRNA-mediated loss of m$^6$A increases nascent translation in glioblastoma

John P. Zepecki[1], David Karambizi[1], J. Eduardo Fajardo[2], Kristin M. Snyder[3], Charlotte Guetta-Terrier[1], Oliver Y. Tang[1], Jia-Shu Chen[1], Atom Sarkar[4], Andras Fiser[2], Steven A. Toms[5], Nikos Tapinos[1,5,6,7] *

1 Laboratory of Cancer Epigenetics and Plasticity, Brown University, Rhode Island Hospital, Providence Rhode Island, United States of America, 2 Department of Systems and Computational Biology, Albert Einstein College of Medicine, Bronx, New York, United States of America, 3 University of Minnesota, College of Veterinary Medicine, St. Paul, Minnesota, United States of America, 4 Department of Neurosurgery, Drexel Neuroscience Institute, Philadelphia Pennsylvania, United States of America, 5 Department of Neurosurgery, Brown University, Providence Rhode Island, United States of America, 6 Cancer Biology Program, Brown University, Lifespan Cancer Institute, Providence RI, USA, 7 Carney Institute for Brain Science, Brown University, Providence, RI, USA

* nikos_tapinos@brown.edu

**Data Availability Statement:** Data generated for this study are available through the Gene Expression Omnibus (GEO: GSE114222).

**Funding:** N.T. acknowledges funding from National Cancer Institute under grant number

## Abstract

Within the glioblastoma cellular niche, glioma stem cells (GSCs) can give rise to differentiated glioma cells (DGCs) and, when necessary, DGCs can reciprocally give rise to GSCs to maintain the cellular equilibrium necessary for optimal tumor growth. Here, using ribosome profiling, transcriptome and m$^6$A RNA sequencing, we show that GSCs from patients with different subtypes of glioblastoma share a set of transcripts, which exhibit a pattern of m$^6$A loss and increased protein translation during differentiation. The target sequences of a group of miRNAs overlap the canonical RRACH m$^6$A motifs of these transcripts, many of which confer a survival advantage in glioblastoma. Ectopic expression of the RRACH-binding miR-145 induces loss of m$^6$A, formation of FTO/AGO1/ILF3/miR-145 complexes on a clinically relevant tumor suppressor gene (CLIP3) and significant increase in its nascent translation. Inhibition of miR-145 maintains RRACH m6A levels of CLIP3 and inhibits its nascent translation. This study highlights a critical role of miRNAs in assembling complexes for m$^6$A demethylation and induction of protein translation during GSC state transition.

## Author summary

Cellular plasticity and epigenetic adaptation of human glioblastoma stem cells to the tumor microenvironment is a hallmark of this devastating disease. With our present work, we discover the relationship between miRNAs and the RNA methylation machinery in human glioblastoma and show how miRNA-induced loss of m6A results in increase in protein translation of clinically important transcripts during glioblastoma stem cell differentiation. Leveraging the dynamic functions of these miRNAs can be important in the design of optimal therapeutics targeted at cancer cell plasticity.

R21CA235415. N.T and S.A.T acknowledge Warren Alpert Foundation Grant. The funders had no role in study design, data collection and analysis, decision to publish, or preparation of the manuscript.

**Competing interests:** The authors have declared that no competing interests exist.

## Introduction

$N^6$-methyladenosine (m$^6$A) is the most prevalent internal modification in eukaryotic messenger RNA and depends on methyltransferases for reversible m6A post-transcriptional installment within the consensus sequence of G(m6A)C (70%) or A(m6A)C (30%)[1,2]. RNA methylation studies in human cells and mouse tissues show m6A enrichment within long exons and around stop codons[3,4], suggesting a fundamental regulatory role of m6A modifications in gene expression. Recent seminal findings show that m6A placement on a given transcript significantly contributes to its fate, marking it for degradation or stabilization and ultimately impacting translation outcome[5]. Such discoveries have led to increased interest in m6A regulation as it relates to various disease processes.

Glioblastoma (GBM) is the most prevalent primary brain tumor. Current treatments include surgical resection followed by radiation and chemotherapy[6]. Even with this multi-therapeutic approach, tumor recurrence is inevitable[7]. A population of glioma stem cells (GSCs) within the tumor mass are believed to be responsible for the aggressiveness, migratory propensity and therapeutic resistance observed in GBM[8–11]. Additionally, GSCs exhibit remarkable plasticity, are able to transition between immature and differentiated stages and can reversibly express various phenotypic markers in response to changes in the tumor micro-environment[12,13]. GSCs are characterized by a network of DNA mutations[14], chromosomal fusions[15], DNA methylation patterns[16] and intratumoral heterogeneity[17]. Moreover, GSCs self-renewal and tumorigenicity properties have also been linked to m$^6$A RNA and m$^6$A writers such as Mettl3/14[18,19] and erasers such as FTO and ALKBH5[19,20]. However, the role of epigenetic mechanisms influencing post-transcriptional mRNA modifications during the transition of GSC to differentiated cell has not been adequately addressed. Here, we integrate RNAseq, meRIP-seq and ribo-seq in stem and differentiated cells from three different patient derived samples representing defined GBM subtypes (mesenchymal, proneural and classical). Using these integrated data, we perform an unsupervised comparative analysis between stem and differentiated cell states in order to uncover how m6A changes relate to translation during the process of cellular differentiation.

Comparative analysis reveals a pattern of significant m6A loss in transcripts that are most efficiently translated following differentiation. Focusing on transcripts with this conserved pattern across all three subtypes, we note a significant enrichment for miRNAs binding motif within the RRACH sequences of such transcripts. Functional testing of relevant RRACH binding miRNAs shows miRNA mediated transcript specific demethylation, but with no actual change in transcript expression. Additionally, expression of these microRNAs corresponds with increased FTO-transcript association and increased RNA demethylase activity. In order to further interrogate the mechanism, we concentrate on a clinically relevant, tumor suppressive transcript (CLIP3). We discover that miR-145 mediates the formation of an FTO-AGO1 complex on the transcript, culminating in CLIP3 m$^6$A demethylation and corresponding increase in nascent translation.

## Results

### Characterization of GSCs and differentiated cells

GSCs were collected from three patients in an IRB approved protocol (Geisinger Medical Center, Danville, PA). The GSCs, denoted GBM1, GBM2 and GBM3 represent different categories of the TCGA-based Verhaak classification scheme, which categorizes GBM into three subtypes based on transcriptional features[21]. GBM1 corresponds to the mesenchymal subtype, GBM2 belongs to the proneural subtype, and GBM3 belongs to the classical subtype. To characterize

differences between the three cell types and validate the 7-day differentiation process from GSCs to differentiated glioma cells (DGCs), we performed transcript expression analysis for GSC and differentiated cell markers. GSCs initially expressed known markers of stemness, including oligodendrocyte transcription factor (*Olig2*), *Sox2* and prominin-1 (*CD133*)[22], but lacked expression of glial fibrillary acidic protein (*GFAP*), a marker of differentiated glial cells in culture[23] (Fig A in S1 Text). However, following differentiation, all lines of GSCs downregulated *Olig2* and *CD133* and exhibited an increase in expression of *GFAP*. Moreover, all GSC lines demonstrated self-renewal potential during limiting dilution assays and exhibited tumor-forming ability following transplantation into immunocompromised mice (Fig A in S1 Text).

## Translation, m6A RNA methylation, and transcriptome profiling of GSCs and differentiated cells

We conducted ribosome profiling and m6A RNA sequencing analyses in GSCs and DGCs isolated from three patients as depicted in Fig 1A. To validate that the m6A antibody enriched for m6A RNA during immunoprecipitation, we performed dot blot of the immunoprecipitated RNA versus input RNA in GSCs and differentiated progeny (Fig A in S1 Text). Similarly, for Ribo-seq validation we show the ribosome profile and ribo-seq reads distribution and median between stem and differentiated progenies showing that median values and read distribution are reproducible across experiments (Fig A in S1 Text). To quantify translation efficiency (TE), we calculated the base-2 logarithm of the ratio of mRNA expression levels in the polysome fraction to that of total mRNA per transcript. A comparison of TE between GSCs and their respective DGCs shows a statistically significant increase in median TE with differentiation irrespective of GBM subtype (p<0.001 for GSCs to DGCs1/2/3, Wilcoxon test) (Fig 1B).

In order to characterize the general m6A peak profile in GSCs and DGCs, the total peak number and corresponding total transcripts/genes were quantified in each group. Approximately ten thousand peak regions were identified per sample pertaining to 3401–5796 genes (Fig 1C, upper panel). The location of the methylation events in the mRNA follows the general distribution previously reported, with notable enrichment around termination codons (Fig B in S1 Text). Although the mean m6A peak number per transcript increased with differentiation in two of the GSC lines, consistent with prior findings[19], one of the subtypes showed no change (Fig 1C, lower panel).

To identify the relation between m6A change and transcript abundance between GSCs and DGCs we quantified the change in m6A levels per transcripts per sample by calculating the difference in total m6a per transcript between both differentiated and stem cells. Transcript abundance was quantified by obtaining the fold change between both cell states. Pearson correlation was subsequently conducted on all GSCs. This shows that correlation between transcript abundance and m6a levels may be GSC dependent as GSC1 shows a trend opposite of GSC2 and GSC3. GSC1 exhibit a negative correlation between m6a and transcript abundance, while GSC2 and GSC3 show the opposite (Fig B in S1 Text).

To determine the general link between RNA methylation, transcription and TE in GSCs and DGCs, we performed an integrated comparative analysis on cellular state (stem vs. differentiated) independent of subtype. First, transcript levels and TEs were obtained in GSCs (n = 3) and DGCs (n = 3). DEseq2 and Ribodiff were used to calculate RNA expression and TE fold change between GSCs and DGCs. Second, to quantify the m6A change between stem and differentiated cell states, the average m6A for GSCs was subtracted from the average m6A for DGCs to obtain the mean peak difference per transcript.

Transcript changes in m6A RNA, transcription levels and TE during differentiation were integrated and depicted on a 3D graph the z, x and y axis respectively representing changes in

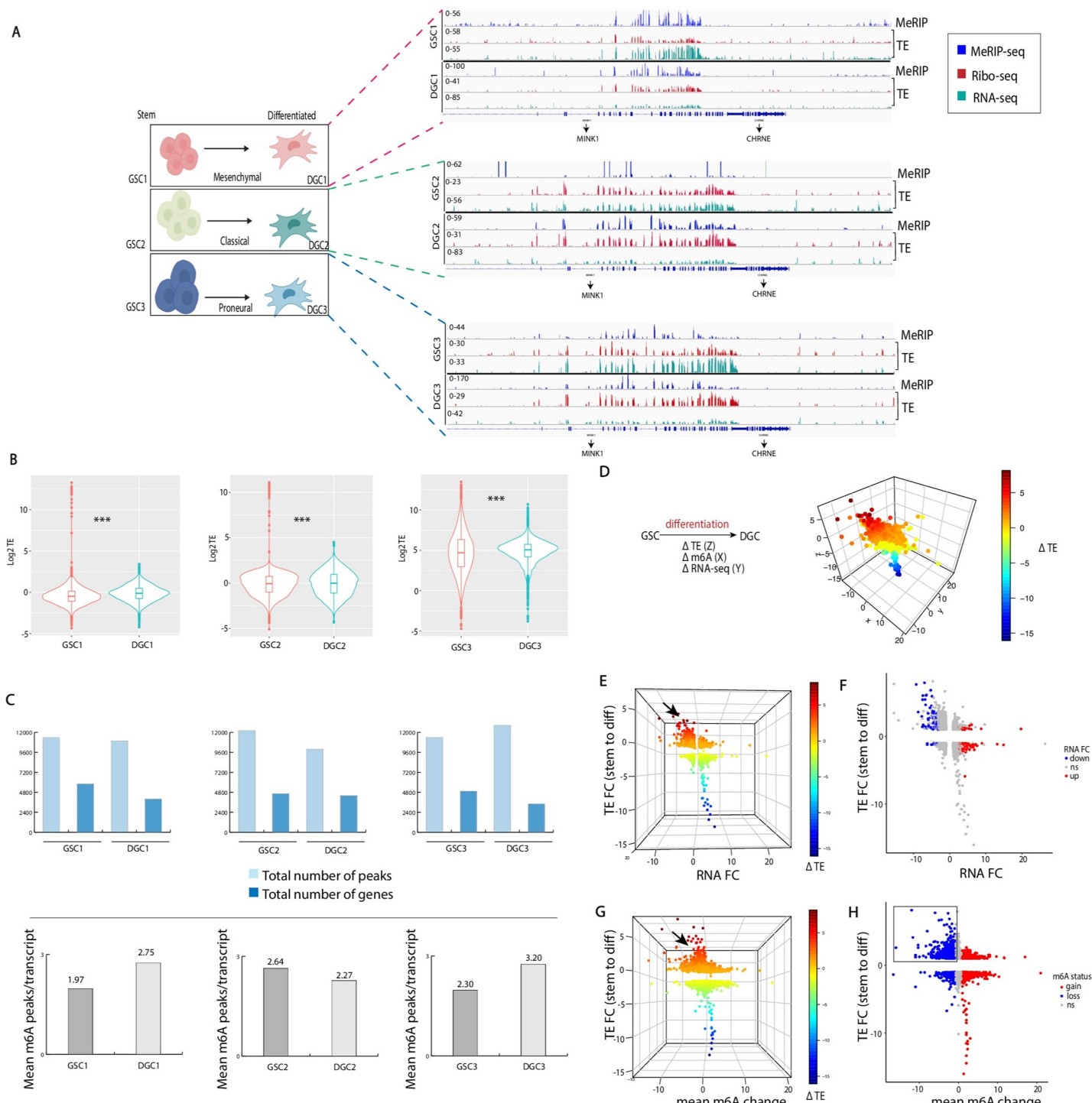

**Fig 1. Translation, methylation and transcriptome profile of GSCs and differentiated progenies.** A) MeRIP-seq and RNA-seq normalized Ribo-seq peak profiles representative GSCs subtypes (mesenchymal, classical and proneural) and differentiated progenies (DGCs) (GSC, n = 3; DGC, n = 3). Specific sequence region shown for illustration of peaks profile across cells. B) Median translation efficiency for GSCs and differentiated progenies. Translation efficiency (TE) derived from the base-2 logarithm of the ratio of mRNA expression levels in the polysome fraction to that of total mRNA per transcript (***p<0.001 for GSCs to DGCs1/2/3, Wilcoxon test). C) Quantification of number of peaks in GSCs and DGCs per subtype. Upper panel shows total number of peaks and associated number of genes per sample; lower panel indicates average peak per transcript in GSCs and corresponding DGCs. D) Change in methylation, m6A and TE following differentiation shown in 3D plot and gradient intensity represent degree of TE FC; z-axis = TE FC, x-axis = mean m6A change, y-axis = RNA-seq FC. E) Plot rotated along RNA FC and TE FC axes, black arrow indicates region with highest TE FC (gradient intensity represent degree of TE FC; z-axis = TE FC, x-axis = mean m6A change, y-axis = RNA-seq FC). F) 2D scatterplot along the RNA FC and TE FC plane axes which emphasizes upregulated (red, 102 genes) and downregulated subset of genes (blue, 120 genes) following GSCs

to DGCs differentiation (P <0.05, FC = 2). G) Plot rotated along mean m6A change and TE FC axes, black arrow indicates region with highest TE FC (gradient intensity represent degree of TE FC; z-axis = TE FC, x-axis = mean m6A change, y-axis = RNA-seq FC). H) 2D scatterplot along the mean m6A change and TE FC axes shows transcripts demarcated by average gain and loss of m6A; cut-off of $1 \leq$ and $-1 \geq$ respectively.

TE, m6A, and RNA levels respectively (Fig 1D). Here, a given transcript's gradient intensity is dependent on its $\Delta$ TE. Generally, the highest TE signal localizes to the left upper quadrant, representing transcripts that have incurred a positive change in TE and a negative change in methylation during differentiation (Fig 1D, 1E and 1G).

To further evaluate how transcription and m6A changes related to TE, we performed an extraction of the 3D graph and separately did an in-depth evaluation of the RNA fold-change (FC) vs. TE and the m6A change vs. TE. Transcriptome analysis reveals no dramatic changes in transcript levels during differentiation. 102 genes are upregulated, while 120 genes are downregulated with approximately 98% of the genes showing no statistically significant change in expression (Fig 1E and 1F). However, m6A analysis in relation to TE during differentiation produces striking results. First, individual transcripts assume all possible outcomes in terms of their distribution (some with m6A loss accompanied by TE decrease or increase and others with m6A gain accompanied by TE decrease or increase), thus reflecting the biological complexity and nuances inherent to the TE/m6A interface (Fig 1H). Second, we identified transcripts that have lost and gained at least 1 m6A peak on average (n = 1382, n = 1455, respectively). We assessed the TE of these two groups of transcripts in all samples (stem and differentiated progeny). Median increase in TE was determined in transcripts that have lost or gained m6A, and significance was calculated with Wilcoxon test. Transcripts that lose m6A during differentiation show highly significant increase in TE in glioblastoma samples irrespective of subtype (GSC1: p<2.2e-16, GSC2: p<3.4e-11, GSC3: p<2.2e-16) (Fig B in S1 Text).

## Transcripts with significant loss in m6A have increased translation efficiency during GSC to DGC transition

To closely investigate the link between m6A RNA and TE during GSC differentiation, 1) we obtained transcript percentile ranking based on transcripts change in TE between each GSC (n = 3) and corresponding DGC (n = 3) (so that transcripts with greatest increase in TE rank highest and those with the most significant decrease in TE rank lowest), 2) we linked each transcript's change in TE to its respective m6A changes, 3) we identified any remarkable and consistent trend(s) in the m6A/TE interface and 4) finally, we identified a set of transcripts with a common trend in all 3 subtypes for further evaluation (Fig 2A).

Change in TE percentile rank plotted against change in methylation in all 3 GSCs and progenies indicates a median methylation change of 0 for 60% of the transcripts, suggesting that the majority of transcripts do not incur significant changes in methylation during differentiation (Fig 2B). However, at the 60$^{th}$ percentile change in TE, the median methylation change precipitously shifts towards negative values (loss in m6A peaks). Analyzed collectively for all GSCs and DGCs, the top 40% transcripts show a statistically significant m6A decrease (P < 0.001, Wilcoxon test) and a marked increase in median TE (P < 0.001, Wilcoxon test) with differentiation (Fig C in S1 Text). Interestingly, an increasing number of m6A peaks are lost as transcripts undergo a greater degree of increase in TE, suggesting a potential incremental link between m6A loss and TE increase (Fig 2B). Furthermore, transcripts experiencing average peak losses greater than 75% show an increase in TE that is significantly higher than those with less than 75% m6A peak loss (Fig C in S1 Text). Similar findings have been previously reported[24]. This suggests that a 75% m6A loss cut-off may generally distinguish transcripts with the most pronounced increase in TE during differentiation.

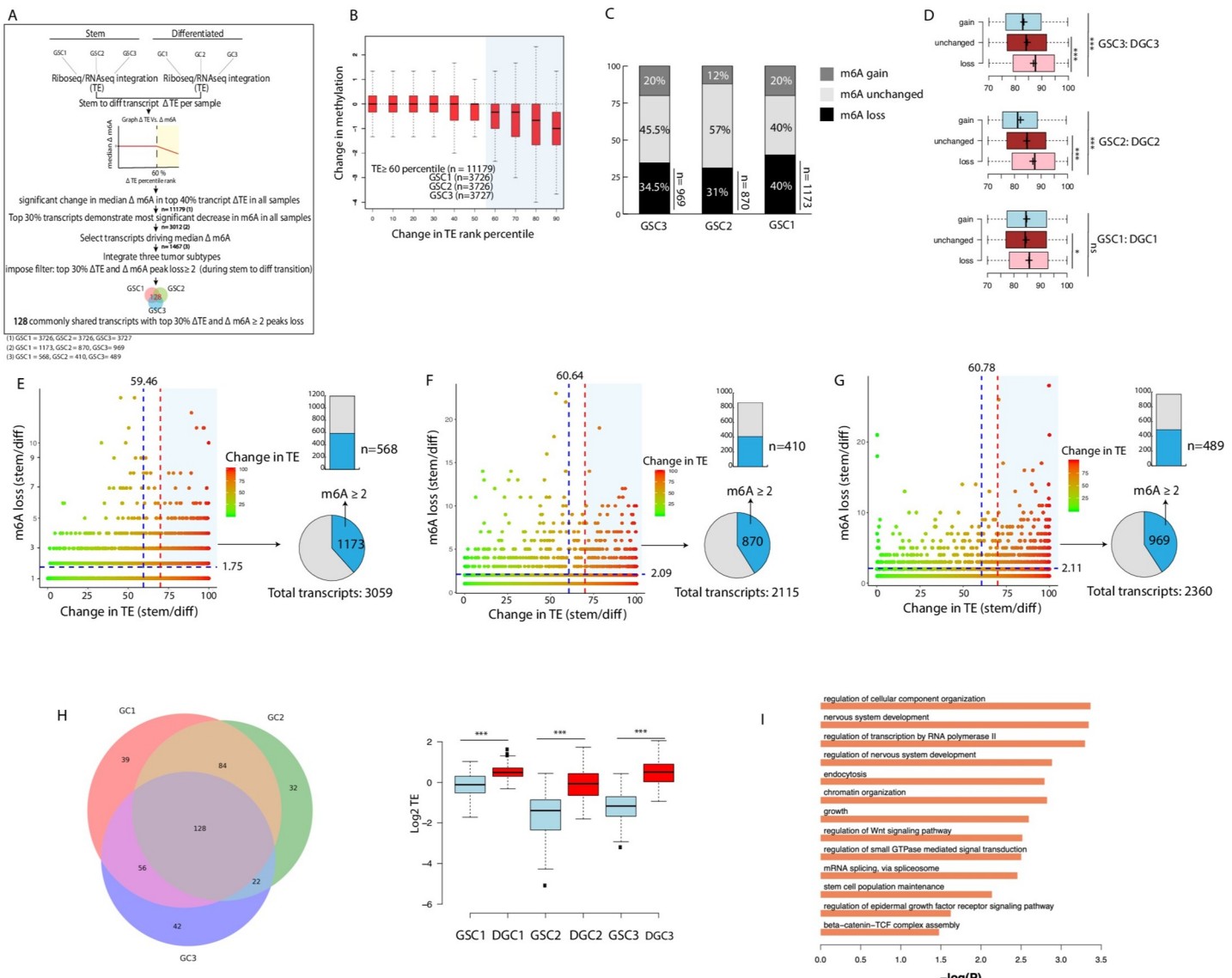

**Fig 2. Transcripts with increased translation efficiency during GSC to DGC transition have a significant loss in m6A methylation.** A) A schematic of the analytical approach with an emphasis on the filtering process from n = 11,179 transcripts to 128 common transcripts. B) Change in TE rank percentile plotted as function of methylation change. Highlight identifies regions with median m6A change ≠ 0, corresponds to GSC 1,2 and 3 (n = 11179) and to top 40% transcripts with the highest increase in TE following differentiation. Change in TE rank obtained by measuring degree of change in TE of individual transcripts between their respective GSC and DGC state and then ranking the transcripts accordingly (ranges from highest rank: greatest increase in TE with transition; to lowest rank: most decrease in TE). C) Percent composition of top 30% transcripts with most significant change in TE based on m6A status in individual samples (gain/unchanged/loss). D) Change in TE rank in transcripts grouped by m6A status (gain/unchanged/loss) (***p<0.001, Wilcoxon test). Crosses represent mean values. E) GSC1: Transcript m6A loss binning correlated with change in TE, vertical blue line indicates median change in TE rank, vertical red line demarcates the top 30% change in TE ranks, horizontal blue lines represent average m6A peak loss across all peaks. Pie chart shows top 30% transcripts as a fraction of all transcripts with m6A loss. Stacked graph emphasizes transcripts with ≥ 2 peaks loss from the top 30% ranked transcripts in terms of TE change during differentiation. F) GSC2: see E. G) GSC3: see E. H) Shows 128 common transcripts across all samples, extracted from transcripts with ≥ 2 peaks loss and change in TE rank ≥ 70th percentile. Boxplot showing log2 TE in individual samples in respective GSCs and DGCs pairing (***p<0.001, Wilcox test). I) Pathway enrichment of 128 common transcripts across subtypes.

Since the trend in m6A loss becomes even more pronounced at the 70th percentile, we elected to collect the top 30% most efficiently translated transcripts per sample for more in-depth analysis. Of these selected transcripts, the majority of transcripts (40–57% in the three GSC lines) had no change in m6A (Fig 2C). However, transcripts with m6A peak losses show

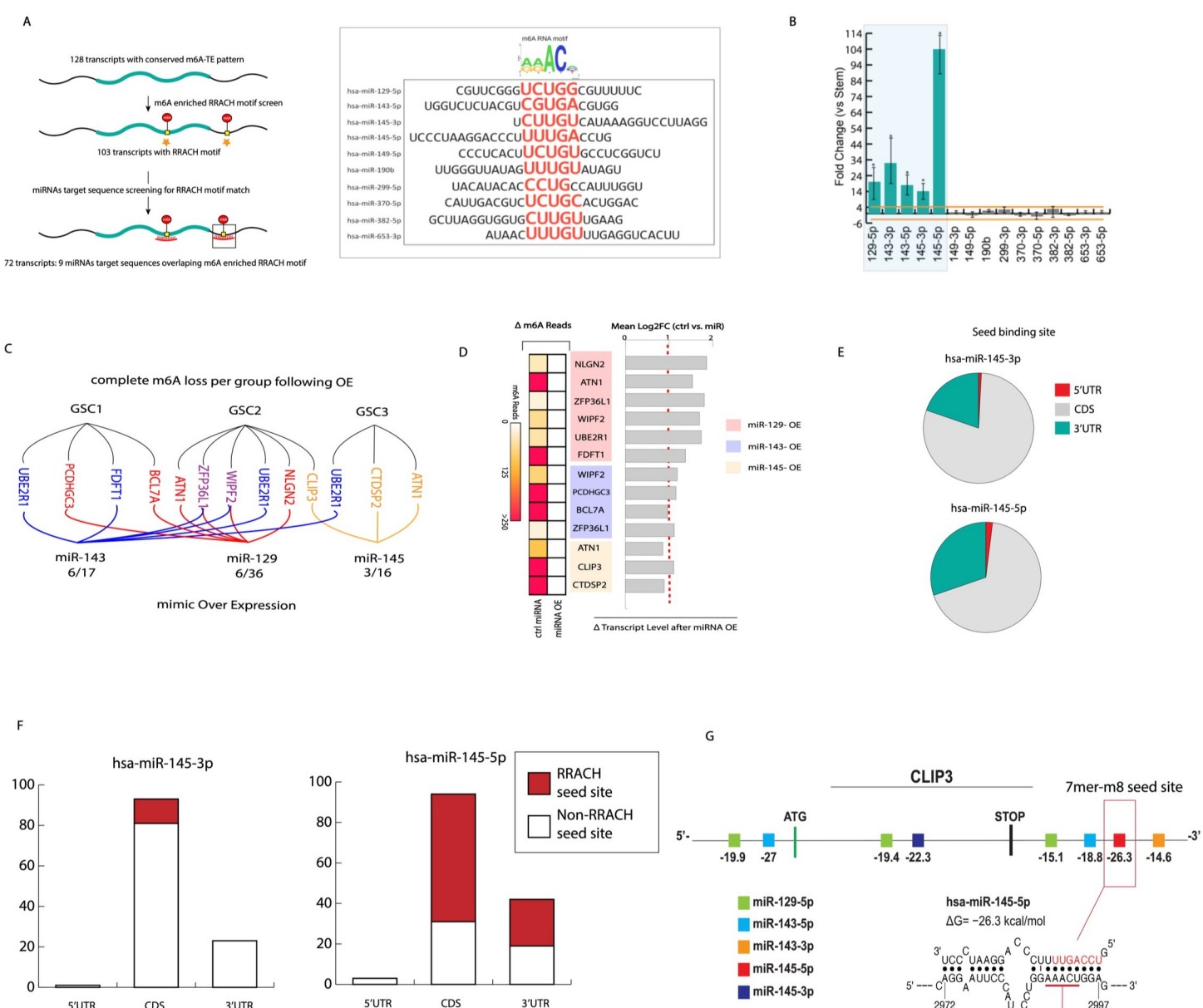

**Fig 3. Ectopic expression of miRNAs predicted to bind within m6A enriched RRACH motifs results in target transcript demethylation without corresponding downregulation.** A) Schematic of the identification process for the 9 miRNAs that are complementary to m6A enriched RRACH motifs. The 72 target transcripts have lost methylation and increased TE during cellular transition across all samples. B) qPCR of all 9 candidate miRNAs. Fold change represents the difference in miRNAs expression between GSCs and differentiated progenies (used fold change of 4 for significance cut-off). C) Ectopically expressed miRNA mimics in respective GSCs and their target transcripts. D) Transcripts methylation and expression status following over expression of miRNA mimics. E) Predicted fraction of miRNA 145-3p and 5p seed binding sites by transcript regions. All binding sites are from sets of 72 transcripts targeted by miRNA 145 (145-3p: 1 site in 5'UTR/ 93 sites in CDS/23 sites in 3'UTR; 145-5p: 3 sites in 5'UTR/ 94 sites in CDS/ 42 sites in 3'UTR). Predictions from human V-Clip based S fold. F) Illustrates fractions of miRNA 145 seed sites predicted to bind within RRACH motifs over all predicted seed binding sites. (145-3p: (0/1) site at 5'UTR/ (12/94) or 13% of sites at CDS/(0/23) at 3'UTR; 145-5p: (0/3) sites at 5'UTR/ (63/94) or 67% of sites at CDS/(23/42) or 55% of sites at 5'UTR). G) S fold derived depiction of predicted binding sites of miRNA 143–145 cluster and miRNA 129-5p on CLIP3 transcript.

the greatest increase in TE relative to transcripts with m6A gain or without change (Fig 2D). In addition to the 70ᵗʰ percentile cut-off, we imposed a methylation cut-off of 2 peak loss in order to capture all transcripts significantly fitting the m6A loss and increase in TE trend. This

corresponds to ~20% of all transcripts with m6A loss in each sample (n = 568, n = 410, n = 489; GSC1, 2, 3 respectively) (Fig 2E, 2F and 2G; Fig C in S1 Text).

Lastly, we identified 128 transcripts, common to all samples, that have lost at least 2 m6A peaks during differentiation and are in the top 30% for most efficiently translated transcripts (Fig 2H). These 128 transcripts 1) show significant increase in log2 TE in all three samples following differentiation (Fig 2H), 2) do not experience a statistically significant change in transcript levels with differentiation as more than 90% remain unchanged (Fig C in S1 Text); and 3) are enriched for key pathways such as Wnt and beta-catenin signaling (Fig 2I).

## miRNA target sites are enriched within the m$^6$A peaks of demethylated transcripts

Based on our observed general link between methylation loss and increase in translation rate, we hypothesized that a regulatory mechanism may exist that connects these two processes. We would like to clarify that throughout the manuscript we consider loss of m6A as demethylation since we are monitoring the m6A status of the same transcripts between GSCs and their differentiated progeny and we provide evidence for the role of the RNA demethylase FTO on this mechanism (Figs 4 and 5). It is not possible with the present analysis to determine if nascent transcripts are generated with *de novo* absence of m6A.

We investigated if changes in translation could be ascribed to altered RNA stability during GSC differentiation. We performed RNA-seq combined with RNA Pol II ChIP-Seq in GSCs and DGCs. RNA Pol II ChIP-Seq measures binding of RNA Pol II across the genome, providing information on levels of nascent transcription [25]. These assays were spike normalized to enable accurate quantification and comparisons between samples. RNA Pol II ChIP-Seq signals across the upregulated genes from RNA-seq revealed that these genes were more highly transcribed in GSCs (Fig D in S1 Text). Likewise, downregulated genes were less actively transcribed. These data indicate that there is no evidence for RNA stability changes during differentiation of GSCs. Second, we determined the status of the methylation machinery following differentiation and found that mRNA levels of pertinent m6A writers/erasers and readers remain largely undisturbed (Fig E in S1 Text). Third, protein levels of FTO, AlkBH5, Mettl3 and Mettl14 do not show any difference during GSC differentiation suggesting the observed loss of m6A during GSC differentiation is not due to decreased expression of m6A erasers or writers (Fig E in S1 Text). These data suggest that observed methylation alterations might likely not be due to changes in transcriptomic or proteomic levels of the m6A machinery itself.

It has been shown that m6A peak regions are highly enriched for miRNA binding and that miRNAs can orchestrate the association of an m6A writer (METTL3) to target transcripts [26]. We looked to investigate whether miRNAs could be implicated in the observed trend of m6A loss potentially via the recruitment of transcripts modifying methylation modulators. To shed light onto some aspects of this potential process, the top transcripts with the top 30% increased TE during differentiation were collected and grouped into 128 common transcripts and others (transcripts in the top 30% but that did not follow the m6a/TE trend across all GSCs). Wilcoxon test was performed on the change in TE percentile of the 128 common transcripts versus other top 30% non-common transcripts across all GSCs. The 128 common transcripts experience the greatest increase in TE amongst the top 30% most efficiently translated transcripts (Fig D in S1 Text). Next, we determined the fraction of transcripts with m6a loss and increase in TE whose miRNA binding sequence overlaps a RRACH motif, which has been identified as the canonical signal for m$^6$A RNA methylation [27,28] and matched motif genomic locations with those of the predicted target sequences of all human miRNAs in the micro-rna.org collection [29] (Fig 3A) (Fig D in S1 Text). The RRACH motif was present in the

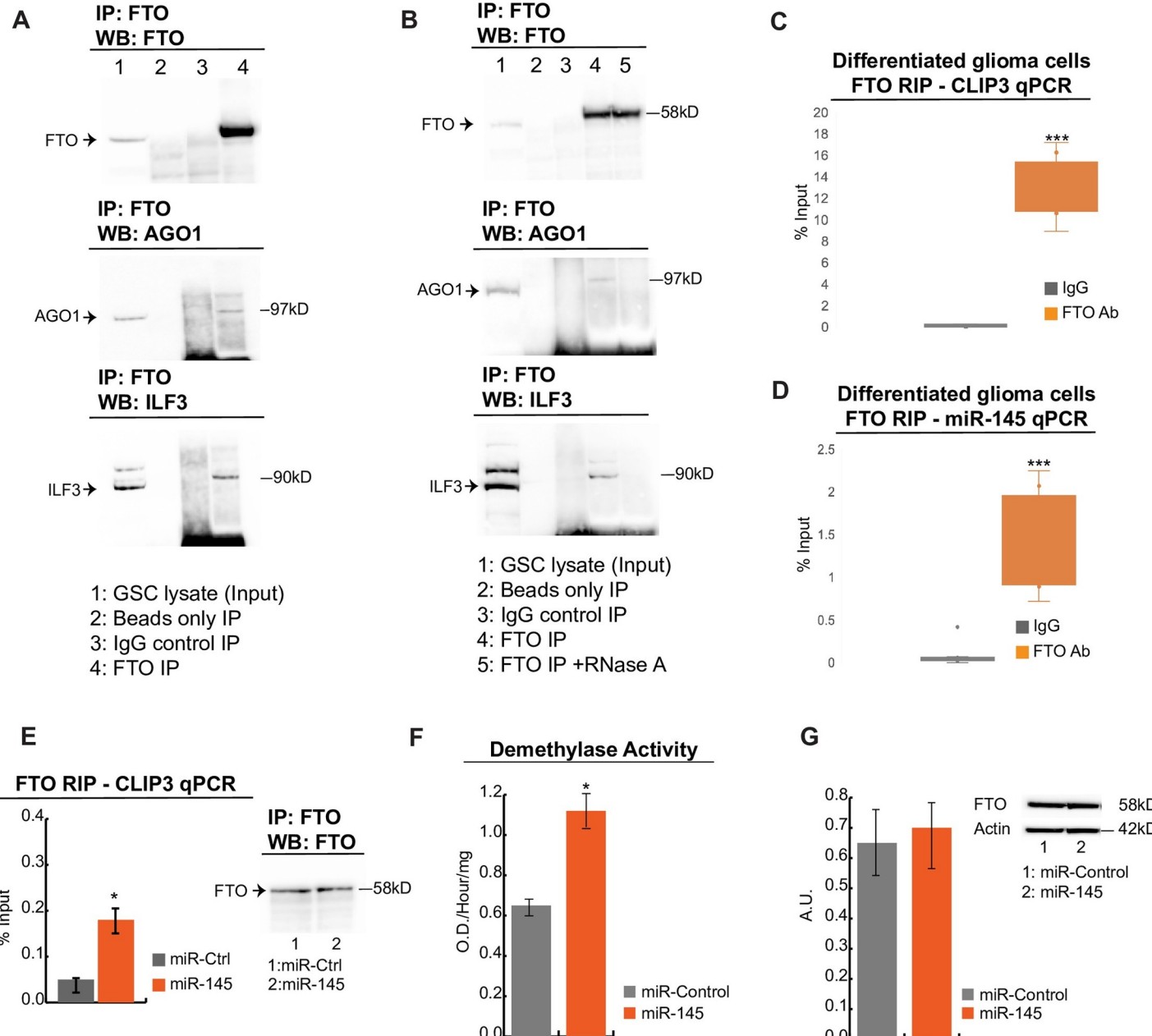

**Fig 4. GSC differentiation induces AGO1/FTO/miR-145 complex formation on CLIP3 mRNA and increased FTO demethylase activity.** A) Protein lysates of DGCs (lane 1) were immunoprecipitated with FTO antibody (lane 4), isotype matched IgG (lane 3) and Protein A beads (lane 2) as control. Western blot was performed using FTO, AGO1 and ILF3 antibodies, which showed that FTO interacts with both AGO1 and ILF3 in DGCs. Immunoprecipitations were performed in 2 biological replicates. B) Protein lysates of DGCs (lane 1) were immunoprecipitated with FTO antibody in the absence (lane 4) or presence of RNase A (lane 5), isotype matched IgG (lane 3) and Protein A beads (lane 2) as controls. Western blot was performed using FTO, AGO1 and ILF3 antibodies, which shows that the FTO/AGO1/ILF3 complex is RNA-dependent. C) qPCR detection of CLIP3 mRNA immunoprecipitated in complex with FTO/AGO1/ILF3. Results are presented as percent of input sample compared to non-specific immunoprecipitation using isotype-matched IgG (***p<0.001, Student's t-test). D) qPCR detection of miR145 immunoprecipitated in complex with FTO/AGO1/ILF3 and CLIP3 mRNA. Results are presented as percent of input sample compared to non-specific immunoprecipitation using isotype-matched IgG (***p<0.001, Student's t-test). E) qRT-PCR detection of *CLIP3* following FTO RNA immunoprecipitation (RIP) after transfection of GSCs with non-targeting miRNA (miR-ctrl) or miR-145. miR-145 induces significant increase in binding of FTO to CLIP3 mRNA. Results are presented as average +/- SD from three independent biological replicates. Significance was calculated with a two-way paired Student's t-test (*p<0.05, df = 4) Western blot insert shows FTO WB following FTO IP on miR-Ctrl and miR-145 transfected cells. F) Transfection of GSCs with miR-145 results in significant increase in cellular demethylase activity. Demethylase activity was calculated with a colorimetric assay using synthetic m6A RNA as substrate and the results are presented as average +/-SD from three independent experiments. Significance was calculated with an unpaired Student's t-test (*p<0.05, df = 3). G) Transfection of miR-145 in GSCs does not change the protein expression levels of FTO. Western blots were repeated three independent times and the results are presented as average +/- SD. Densitometric quantification was performed using ImageJ.

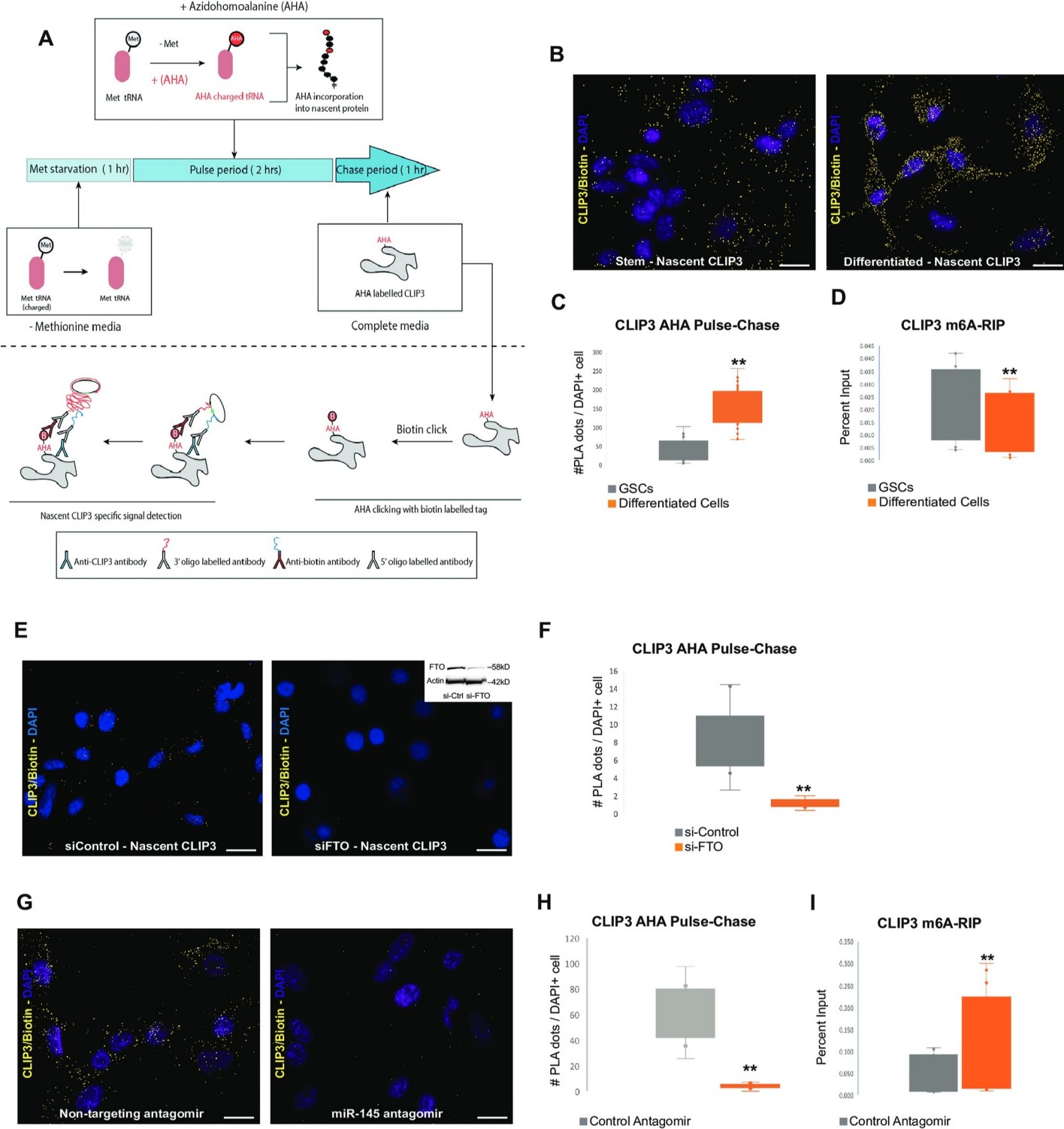

**Fig 5. miR-145 induces m6A loss and nascent translation of CLIP3.** A) Schematic representation of the AHA pulse chase experiment to detect nascent translation of CLIP3 through Click-mediated Biotin labeling of AHA-incorporated nascent transcripts followed by PLA detection using Biotin and CLIP3 specific antibodies. B) Differentiation of GSCs induces nascent translation of CLIP3. Representative images of AHA Pulse-Chase to detect CLIP3 nascent translation in GSCs (left panel) and after differentiation (right panel). Positive CLIP3 PLA dots were pseudo-colored yellow and DAPI nuclei purple. Scale bar: 100μ. C) Quantification of number of PLA dots per cell shows that nascent translation of CLIP3 is significantly increased after differentiation of GSCs. Significance was calculated from at least 200 cells per

condition using a Student's t-test (**p<0.0005, n = 3 biological replicates). D) m6A-RIP for *CLIP3* in GSCs and after induction of differentiation, shows that transition to differentiation results in significant m6A demethylation of *CLIP3*. Significance was calculated from n = 4 biological replicates (**p<0.002, Paired Student's t-test). E) Representative images of AHA Pulse-Chase to detect CLIP3 nascent translation in differentiated cells after transfection of si-Control (non-targeting) or si-FTO. Inhibition of FTO expression rescues the increase in nascent translation of CLIP3. Insert shows Western blot detection of FTO inhibition following siRNA transfection. Actin was used as loading control. F) Quantification of number of PLA dots representing nascent transcripts of CLIP3 per cell in si-Control and si-FTO transfected cells. (**p<2e-5). G) Representative images of AHA Pulse-Chase to detect CLIP3 nascent translation in differentiated cells after addition of a non-targeting miR-antagomir (left panel) or a specific miR-145 antagomir. Inhibition of miR-145 expression by the miR-145 antagomir, inhibits nascent translation of CLIP3. Scale bar: 100μ. H) Quantification of number of PLA dots per cell shows that inhibition of miR-145 expression using a miR-145 antagomir results in significant inhibition of nascent translation of CLIP3. Significance was calculated from at least 200 cells per condition using a Student's t-test (**p<0.0005, n = 3 biological replicates). I) m6A-RIP for *CLIP3* in differentiated glioblastoma cells transfected with a non-targeting antagomir or a miR-145 specific antagomir. Inhibition of miR-145 induces significant increase of m6A methylation of *CLIP3*. Significance was calculated from n = 4 biological replicates (**p<0.02, Paired Student's t-test).

methylated peak regions of 103 of the 128 transcripts. We identified nine miRNAs with target sequences that overlap the motif and verified that each miRNA had a sequence complementary to the RRACH motif with at most 1 nt mismatch, suggesting that the m⁶A peak regions may be targeted by these miRNAs (Fig 3A). 72 out of the 103 transcripts had at least one of the nine miRNAs binding within an m⁶A peak in all three patient samples tested (Fig 3A, Fig E in S1 Text). We subsequently screened these 9 microRNAs by assessing expression levels in glioma stems and differentiated cells. A fold change cut-off of 4 was imposed for significant miRNA expression change. Interestingly, this showed a significant increase in known tumor-suppressive miRNAs following differentiation: the miRNA 143–145 cluster as well as miR-129-5p (Fig 3B)[30,31].

## Expression of miRNAs induces loss of m⁶A RNA from target transcripts

Next, we assessed if these RRACH binding microRNAs played a role in target transcripts methylation status. We ectopically expressed miRNA 143, 145 and 129 mimics in a subtype specific manner because the GSCs preferentially express one of the three miRNAs following differentiation (Fig 3C). We performed MeRIP-seq and RNA-seq on transfected GSCs biological replicates with a focus on the set of the 72 transcripts targeted by each of the three miRNAs. First, target transcripts saw a significant reduction in m6A methylation peaks reads (Fig 3D). Second, RNA-seq and follow up confirmatory qPCR on miRNA targets in transfected cells showed no changes in expression and in some instances showed a slight increase (Fig 3D). Taken together, ectopic expression of these tumor suppressive RRACH binding miRNAs seem to facilitate transcript specific demethylation without engaging in expected miRNA mediated transcript downregulation.

We next sought to investigate a potential link between these miRNAs and translation as well as m6A machinery recruitment. To achieve this, we focused on miRNA 145-5p and the transcript CLIP3 for several reasons: 1) miRNA 145 is a well-documented miRNA that is largely regarded as a tumor suppressor across multiple cancer types[32], 2) miRNA 145-5p experiences a dramatic increase in expression following differentiation far surpassing other candidate miRNAs (Fig 3B) 3) miRNA 145-5p seed region (5' 2–8 nucleotide) shows significant projected binding within the RRACH motif of sets of 72 targets transcripts (70% within CDS and ~55% in 3'UTR regions) (Fig 3E and 3F) 4) although 72 transcripts of interest are enriched in key regulatory pathways, CLIP3 is the only one with consistent clinical significance across all tested TCGA platforms, with elevated expression associated with increased survival (Fig E in S1 Text), and 5) 145-5p and CLIP3 have a predicted strong interaction at the 3'UTR, having the lowest Gibbs free energy interaction of all miRNA tested against CLIP3 and showing perfect CLIP3 RRACH motif complementarity at miRNA 145-5p 7mer-m8 seed site (Fig 3G).

## GSC differentiation induces formation of an FTO/AGO1/ILF3/miR-145 complex on CLIP3 mRNA

Since GSC differentiation increases the expression of miR-145 and induces CLIP3 m6A demethylation, we sought to determine how miR-145 interacts with the m6A methylation machinery to mediate transcript demethylation. In general, miRNAs are delivered into the cytoplasm as part of Argonaute (AGO) protein–RNA complexes[33]. A fraction of AGO1 protein does not possess Dicer activity and has been shown to interact with various proteins including Interleukin enhancer binding factor 3 (ILF3)[33], which is a double-stranded RNA binding protein that complexes with other proteins, small noncoding RNAs, and mRNAs to regulate gene expression and stabilize mRNAs[34]. ILF3 has been also shown to interact with FTO[35], so we hypothesized that AGO1 could form multimeric complexes with ILF3/FTO/miRNA on target mRNAs to mediate transcript demethylation. We performed RNA immuno-precipitation with an FTO antibody followed by Western blot and qPCR to detect the presence of protein/RNA complexes following GSC differentiation. We show the formation of FTO/AGO1/ILF3/miR-145/CLIP3-mRNA complex (Fig 4A, 4C and 4D) in DGCs. To determine if the FTO/AGO1/ILF3 complex is RNA-dependent we performed co-immunoprecipitation in the presence or absence of RNase A. We show that the presence of RNA is necessary for the formation of the complex since incubation with RNase A inhibits the interaction between FTO, AGO1 and ILF3 (Fig 4B).

## miR-145 induces binding of FTO to CLIP3 mRNA and increases cellular m6A demethylase activity

FTO efficiently demethylates m$^6$A RNA *in vitro* and the cellular levels of m$^6$A RNA are affected by the enzymatic activity of FTO *in vivo* [36]. We performed RIP to precipitate endogenous FTO and its associated mRNAs from three patient-derived GSCs, after overexpression of miR-145 mimics. This showed that overexpression of miR-145 results in significant increase in the association of FTO with CLIP3 mRNA in GSCs (Fig 4D). To show that the recruitment of FTO is not specific to miR-145 but rather a mechanism that RRACH-binding miRNAs employ for regulation of their targeted mRNAs, we performed FTO RIPs after overexpression of the other two tumor-suppressive miRNAs miR-143 and miR-129 (Fig 3B) followed by qPCR of their target genes UBE2R2 and ATN1 respectively. This showed that expression of these RRACH-binding miRNAs significantly increases the association of FTO with UBE2R2 and ATN1 mRNAs (Fig E in S1 Text).

Next we determined if the increased association of FTO with CLIP3 induced by miR-145, results in enhanced functional demethylase activity of FTO. We performed a colorimetric demethylase assay, which demonstrated that overexpression of miR-145 induces a significant increase in cellular demethylase activity in lysates of GSCs (Fig 4E) without affecting total FTO protein levels (Fig 4F). In conjunction, our data suggest that GSC differentiation induces Ago1-mediated delivery of miR-145 to CLIP3 mRNA in complex with FTO and that the increased presence of miR-145 induces the activity of FTO towards m6A demethylation.

## GSC differentiation induces miRNA-dependent nascent translation and m$^6$A demethylation of CLIP3

To determine the effect of the miR-145 induced m6A demethylation on protein synthesis of CLIP3 during differentiation of GSCs, we performed pulse-chase experiment in individual cells using L-Azidohomoalanine (AHA) incorporation to quantify nascent protein synthesis [37]. To quantify the amount of AHA modified nascent CLIP3 in GSCs and differentiated

cells, we performed proximity ligation assay (PLA) using antibodies against CLIP3 and Biotin (Fig 5A). This showed a significant increase in nascent translation of CLIP3 during GSC differentiation (Fig 5B and 5C and Fig F in S1 Text). Next, we quantified the levels of *CLIP3* m⁶A RNA following differentiation of GSCs. m⁶A-RIP followed by *CLIP3* qPCR shows that *CLIP3* transcripts are significantly m⁶A demethylated (Fig 5D). To verify that the induction of nascent translation of CLIP3 during GSC differentiation is FTO-dependent we rescued the m6A demethylation of CLIP3 by siRNA-mediated knockdown of FTO and show that inhibition of FTO expression results in significant reduction of the nascent translation of CLIP3 (Fig 5E and 5F).

To confirm that miR-145 regulates the induction of nascent translation of the m⁶A-demethylated *CLIP3* transcript, we transfected differentiated GCs with either a non-targeting antagomir or a miR-145 antagomir to inhibit the expression of miR-145 (Fig F in S1 Text). Inhibition of miR-145 in differentiated GCs results in complete inhibition of nascent translation of CLIP3 (Fig 5E and 5F) and rescue of *CLIP3* m⁶A-demethylation (Fig 5G) without affecting *CLIP3* transcript expression levels (Fig F in S1 Text), suggesting that miR-145 is necessary to induce m⁶A demethylation and induction of nascent translation of CLIP3. To verify that the effect of miR-145 on m6A RNA methylation and induction of nascent translation is specific for transcripts where miR-145 binds within the m6A peak, like *CLIP3*, we determined the effect of the miR-145 antagomir on nascent translation of YTHDF2, which is a transcript that is expressed in human glioma cells (Fig F in S1 Text), has conserved binding sites for miR-145 at the 3'-UTR (www.targetscan.org) and is not m6A RNA demethylated during differentiation of GSCs. This showed that inhibition of miR-145 significantly induces translation of YTHDF2 (Fig F in S1 Text), suggesting that miR-145 functions as a classical miRNA suppressing translation of transcripts with miRNA binding sites at the 3'-UTR, like *YTHDF2* and that the effect of miR-145 to induce nascent translation of *CLIP3* is m⁶A-related.

## Discussion

Following the first identification of human cancer stem cells in leukemia [38], several groups have shown the existence, isolation and characterization of cancer stem cells in brain tumors including glioblastoma [9,39–41]. GSCs are resistant to standard chemotherapy and radiation therapy [42,43] and thus contribute to disease progression and recurrence. Maintenance of the GSC state or induction of differentiation depends on epigenetic influences at both the transcriptional and chromatin regulation level [44–47]. An increasing body of evidence suggests that RNA is capable of influencing how epigenetic states are established and maintained during development, cell division or cell differentiation [48]. However, the relationship between post-transcriptional modifications of RNA (e.g. m⁶A RNA methylation) and the regulation of cellular differentiation remains unclear. In glioblastoma, RNA methylation patterns in GSCs versus those in differentiated glioma cells are unknown, and how differences in RNA methylation influence GSC differentiation has not been studied. Moreover, the role of epigenetic regulators, such as non-coding RNAs, which could affect m6A RNA methylation patterns in GSC and differentiated glioma cells have not been discovered.

GSCs receive a multitude of signals from the tumor microenvironment and must adapt to altered environmental conditions rapidly. GSCs manifest such dynamic cellular adaptation by altering phenotypic expression and undergoing cellular proliferation or cell differentiation. Cellular adaptation to fluctuating environmental conditions often requires the involvement of rapidly responsive post-transcriptional mechanisms, as regulation via transcript levels tuning alone would be too slow. Factors like the temporal delay between transcription and translation limit the speed and capacity at which proteomes can be adapted by cells solely through altering

transcription [49]. Here we investigated whether alterations at the m$^6$A RNA methylation levels could be one of the post-transcriptional mechanisms that could modulate translation rates in GSCs.

We have shown that GSCs and differentiated glioblastoma cells have distinctive patterns of m$^6$A RNA methylation as well as clear differences in translational responses to such cell state specific methylation patterns. Furthermore, we identify a common group of transcripts that undergo RNA methylation peak losses during cell differentiation in all patient samples tested and show that such losses correlate with increased translational efficiency. Additionally, our data generate new questions regarding the mechanism of miRNA-induced m6A demethylation. A recent study indicated that RNA demethylases FTO and ALKBH5 discriminate their m$^6$A targets based on structural rather than primary sequence properties *in vitro*. Specifically, it was shown that m$^6$A serves as a 'conformational marker' which dynamically regulates the overall conformation of the modified RNA and, consequently, the substrate selectivity of m$^6$A demethylases [50]. In addition, it has been shown that m$^6$A methylation can directly impact the thermodynamic stability and conformations of DNA/RNA [51–53]. The structural influence of m$^6$A is also evident in cellular RNA. Recent work revealed that m$^6$A-modified sites exhibit specific structural signatures, and loss of m$^6$A modifications results in a significant loss of these structural signatures [54]. Since the miRNAs in GSCs bind within the RRACH m$^6$A motif, it is plausible that this binding may alter the stability and structural conformation of these mRNAs, making the m$^6$A sites more accessible for recognition and binding by cellular FTO. Increased accessibility of the m$^6$A sites of the miRNA-targeted transcripts could result in increased association of FTO with these transcripts and, subsequently, transcript demethylation.

Recently, CLIP analysis has shown that there is an Argonaute (AGO) binding site in transcripts within 60% of an m6A RRACH motif [55,56]. Furthermore, the RNA demethylase FTO has been shown to shuttle between the nucleus and the cytoplasm and can interact with targets within both cellular compartments [57]. The role of miRNAs in regulating inhibition of transcript expression, mRNA degradation and inhibition of translation initiation has been well defined over the past decade [58–60]. miRNA target sites are generally located in the 3′ UTR of mRNAs and possess strong complementarity to the seed region [61], which is the main criterion for target-site prediction [29,62]. The canonical effector function of miRNA binding to the target transcript is to direct mRNA degradation and subsequent inhibition of translation. Our results here point to a mechanism that deviates from these conventional miRNA functions.

It has been shown that AGO1 and AGO2 proteins form functional complexes with miRNAs, mRNAs, mRNPs [33,63,64] and are associated with RISC and Dicer activity [33]. However, a certain fraction of AGO proteins does not contain RISC and shows little or no Dicer activity. This fraction of AGO1 can associate with the DZF domain (Domain associated with Zinc Fingers) of the ILF3 protein which has been also detected to interact with FTO [35]. We showed here that in DGCs, FTO forms multimeric complexes with AGO1 and ILF3 with miR-145 and CLIP3 mRNA. It is possible that AGO1 and ILF3 stabilize the miRNA—mRNA complex on RRACH motifs and recruit FTO, which demethylates m6A marks and the demethylated transcript is then translated more efficiently. The presence of the miRNA is the rate-limiting step for this function since inhibition of miR-145 resulted in loss of FTO binding to the targeted transcript and inhibition of translation.

In summary, we present the first functional link between loss of m$^6$A RNA methylation and increased translation in human glioblastoma cells as well as a role for miRNAs in the modulation of m$^6$A RNA demethylation in genes that are most efficiently translated during GSC differentiation. Within the glioblastoma dynamic and plastic cellular niche, GSCs can give rise to

DGCs and, when necessary, DGCs can reciprocally give rise to GSCs to maintain the cellular equilibrium necessary for optimal tumor growth. Here, we uncover a set of miRNAs with the capacity to regulate the epitranscriptome and to induce protein translation during GSC cell state transition. We believe that leveraging the dynamic functions of these miRNAs can be important in the design of optimal therapeutics targeted at cancer cell plasticity.

## Materials and methods

### Ethics statement

The institutional review boards at Rhode Island Hospital and Geisinger Clinic approved the collection of de-identified patient-derived Glioblastoma Multiforme (GBM) tissue. All participants provided written informed consent for the use of glioblastoma tissue for research purposes.

### Cell lines

Primary hCSC spheres were cultured from human glioblastoma samples as previously described [40]. All hGCs used in this study (GBM1, GBM2, GBM3) were authenticated by ATCC using Short Tandem Repeat (STR) analysis. All human primary cells used were between passages 5–10. All cultures were routinely tested for mycoplasma contamination using the LookOut Mycoplasma PCR Detection kit (Sigma). To induce differentiation, CSCs were plated on fibronectin-coated dishes in medium containing 10% serum without bFGF, EGF and Heparin. Cultures were maintained in differentiation media for 7 days.

### RNA sequencing

Next-generation RNA-sequencing was performed using an Illumina HiSeq2500 system. Sequence reads were aligned to the human genome (hg19 build) using *gsnap*. Genomic locations of genes and exons, were extracted from the *refGene.txt* file (http://hgdownload.cse.ucsc. edu/goldenPath/hg19/database/refGene.txt.gz).

### m$^6$A RNA immunoprecipitation and MeRIP-seq

Total RNA was isolated from glioma stem and differentiated samples using Trizol and treated with RiboMinus (Thermofisher) to remove ribosomal RNA. Samples were then fragmented with RNA Fragmentation Reagent (Ambion) and 200ug of RNA was immunoprecipitated with m6A antibody (12ug) (Synaptic Systems, 202–003) and Dynabeads (Thermofisher) at 4˚C overnight. The precipitated RNA was then used to perform RNA-seq using an Illumina-HiSeq2500 sequencer.

### M6A dot blot

GSCs were seeded on tissue-culture treated dishes coated with human fibronectin (Millipore) at a concentration of 10 ug/mL in either stem or differentiation medium for a total of 5 days. Total RNA was isolated and purified using Trizol (Thermo) and RNA IP was then performed on 10 ug samples as described [65] with the Imprint RNA Immunoprecipitation Kit (Sigma) and human m6A antibody (Synaptic Systems).

### Polysome fractionation

Polysome fractionation and sample collection was performed as we have previously described [66].

## Polysome association

Read summarization at the gene level was done for all the genes in Refseq using the bam alignment files and in-house scripts, taking only reads with mapping quality of 20 or greater. The number of raw reads mapping to a gene was standardized to reads per kilobase per million reads (RPKM). After discarding genes with fewer than 2 RPKM in the total-RNA samples, we retained 9314 protein-coding genes for analysis. To determine the relative polysome association of each RNA, we divided the number standardized reads in the polysome sample by those in the total RNA sample and calculated the base-2 logarithm of this ratio.

## miRNA overexpression

Glioma cells were grown as attached on cell-treated vessels coated with 10ug/mL human plasma fibronectin (Millipore, FC010) and transfected with X2 (Mirus Bio) lipid and 25 nM miRIDIAN microRNA Mimic (Dharmacon) in complete media without heparin. Heparin-containing complete medium was replaced after 24 hours, and cells were lysed 48 hours post-transfection using the miRCURY RNA Isolation Kit (Cell and Plant, Exiqon).

## Western blot analysis

Cells were lysed 48 hours post-transfection with 1% SDS and quantified via Pierce Protein Assay (Thermofisher) and a spectrophotometer (Biotek) at 562 nm. Western blots were performed according to the protocols suggested by the producer of each primary antibody and were developed with Radiance chemiluminescent substrate (Azure). Images were taken with an Azure c300 chemiluminescent imaging system and band intensity was quantified using ImageJ. The following primary antibodies were used for western blot analyses: ALKBH5 (Millipore, ABE547), FTO (Millipore, MABE227), and Beta-actin (Sigma, A5441). 25ug of protein was used per lane for all Western blots.

## miRNA isolation and expression

miRNA was isolated from human glioma stem and differentiated cells using the miRCURY RNA Isolation Kit (Cell and Plant, Exiqon) and quantified with a NanoDrop 2000 spectrophotometer. Relative miRNA expression between stem and differentiated samples was assessed via RT-qPCR with the miRCURY LNA Universal RT microRNA PCR system (Exiqon) and a StepOnePlus thermocycler (Applied Biosystems). The following miRNA LNA qpcr primer sets (Exiqon) were tested: hsa-miR-129-5p, hsa-miR-143-3p, hsa-miR-143-5p, hsa-miR-145-3p, hsa-miR-145-5p, hsa-miR-149-5p, hsa-miR-149-5p, hsa-miR-190b, hsa-miR-299-3p, hsa-miR-370-3p, hsa-miR-370-5p, hsa-miR-382-3p, hsa-miR-382-5p, hsa-miR-653-3p, hsa-miR-653-5p, let-7a-5p, 16-5p, 103a-3p, 191-5p, 423-3p, and 423-5p.

## RNA Immunoprecipitations

RNA immunoprecipitations were performed with the Imprint RNA Immunoprecipitation Kit (Sigma) according to manufacturer's instructions and 4ug of the FTO primary antibody (Millipore, ABE552) or Rabbit IgG negative control antibody (Sigma) coupled with Protein A Magnetic beads (Sigma). Subsequent RT-qPCR analysis was performed using the SuperScript III First-Strand Synthesis System for RT-PCR (Thermofisher) and the following primers: ATN1 (F:5'-AATGAGGAGTGGACGGAAGAA-3'; R:5'-CTCCGACCCTGCTTGTTGAC-3', UBE2R2 (F:5'-CCACTAAGGCCGAAGCAGAAA-3'; R:5'-TCGTAAAGCAAATCT-GAGCTGT-3'), and CLIP3 (F:5'-TGCTCCACTATGCGTGCAAA-3'; R:5'-TGAAGCGCGTT-CATGTTGGT-3').

Relative gene expression after miRNA overexpression was assessed using the RT$^2$ First Strand Kit (Qiagen) after total RNA isolation with Trizol (Ambion). The following RT$^2$ qpcr primer assays (Qiagen) were used: ATN1, BCL7A, CLIP3, CTDSP2, FDFT1, GAPDH, NLGN2, PCDHGC3, UBE2R2, WIPF2, ZFP36L1.

## Co-Immunoprecipitations

Co-immunoprecipitations were performed with the Imprint RNA Immunoprecipitation Kit (Sigma) according to manufacturer's instructions and 5ug of the FTO primary antibody (Millipore, ABE552) or Rabbit IgG negative control antibody (Sigma) coupled with Protein A Magnetic beads (Sigma), and protein-protein interactions were inferred via SDS-PAGE. Proteins were eluted from beads by boiling with 1x SDS at 95°C for 5 minutes. The following primary antibodies were used for Western blotting following immunoprecipitation: Ago1 (Cell Signaling, 5053S) and ILF3 (ProteinTech, 19887-1-AP).

## RNase FTO-AGOI RNA co-IP

Glioma cells were seeded on tissue-culture treated dishes coated with human fibronectin (Millipore) at a concentration of 10ug/mL in differentiation medium for 5 days. Cells were subsequently lysed using "mild lysis buffer" (Sigma) supplied in the Imprint RNA Immunoprecipitation Kit (Sigma) and RNA was digested with Monarch RNAseA (NEB) for 1 hour at 4C. RNA/protein complexes were IP-ed with human anti-FTO antibody (Millipore) and immunoblotted with human FTO (Millipore), Ago1 (Cell Signaling) and ILF3 (Thermo) antibodies.

## Identification and transcriptome-wide profiling of m6A RNA methylation sites

m6A profiling of glioma stem samples before and after miRNA overexpression was performed using the Magna MeRIP m6A Kit according to manufacturer instructions. mRNA was isolated from glioma stem and differentiated samples using the Dynabeads mRNA Purification Kit (Thermofisher) and subsequently fragmented with RNA Fragmentation Reagent (Ambion) prior to immunoprecipitation. Total RNA was isolated from identical samples using Trizol (Ambion) and fragmented in the same manner. DNA libraries were then prepared using a custom Qiaseq Targeted RNA panel (Qiagen, CRHS-10308Z-88). cDNA from each RNA sample was assigned molecular barcodes followed by a 2-step PCR amplification with intermittent cleanup between each step via QIAseq beads as described by the manufacturer. PCR products were then quantified and sequenced using MiSeq (Illumina).

## Proximity ligation assay

Protein co-localization was assessed via proximity ligation assay (PLA) using the Duolink In Situ Red Starter Kit (Sigma) and PLA-approved primary antibodies for SC-35 (Abcam, ab11826) and FTO (Abcam, ab126605). Cells were fixed in 4% paraformaldehyde and permeabilized with 0.2% Triton X-100 (Sigma). All other steps followed manufacturer's instructions. Images were acquired using an Olympus FV3000 and associated software. Quantification was performed using ImageJ.

## Demethylase assay

Nuclear enzyme activity was assessed with the m6A Demethylase Activity/Inhibition Assay Kit (Epigenase, P-9013-96). Lysates were prepared using the Total Nuclear Extraction Kit I

(Epigenase, OP-0002) following manufacturer instructions. Lysate concentration was estimated via Bradford Assay (Sigma) and a NanoDrop 2000 (Thermofisher).

## Inhibition of miR-145

Knockdown of miR-145-5p in glioma cells was achieved via X2 lipid transfection (Mirus Bio, MIR-6004) of hsa-miR-145-5p miRCURY LNA miRNA Power Inhibitor (Exiqon, YI04102423-DCA) or miRCURY LNA miRNA Power Inhibitor Negative Control A (Qiagen, YI00199006-DDA) at a final concentration of 50 nM and confirmed by RT-qPCR with the miRCURY LNA Universal RT microRNA PCR system (Qiagen) and a StepOnePlus thermocycler (Applied Biosystems). Ct values for hsa-miR-145-5p (Qiagen, YP00204483) were normalized to expression levels of hsa-miR-423-5p (Qiagen, YP00205624) to determine fold change.

## Nascent translation quantification

Fluorescence non-canonical amino acid tagging followed by proximity ligation assay (FUN-CAT-PLA) was performed according to manufacturer protocols [67] with a few modifications. Click-IT AHA (Invitrogen, C10102) was incorporated into cells at a concentration of 500 uM for 2 hours in methionine-free Neurobasal -A media (custom made by Gibco) followed by a 1 hour "chase" with complete medium containing methionine. Cells were fixed in 4% methanol-free formaldehyde (Thermo, 28906) and permeabilized in .4% Triton X-100 (Sigma, T8787). The Click-iT Cell Reaction Buffer Kit was then used to "click" the biotin alkyne (Thermo, B10185) to the incorporated AHA azide at a final concentration of 25 uM overnight at 4˚C. PLA was carried out using Duolink In Situ Red Starter Kit Mouse/Rabbit (Sigma, DUO92101) with 1:500 biotin primary antibody (Abcam, ab201341) and either 1:500 CLIP3 (Abcam, ab74239) or 1:500 YTHDF2 (Proteintech, 24744-1-AP) primary antibodies overnight at 4˚C. Subsequent detection was performed according to manufacturer's recommendations. In order to quantify the PLA signal, 2D image analysis was performed by applying the connected components labeling algorithm implemented in SciPy's open-source software library (http://www.scipy.org/) to the 50x images originally derived from the RFP channel of the Evos FL Auto microscope (Thermo, AMAFD-1000). First, using the OpenCV image processing library, the images were converted from BGR color-space to grayscale and then a Gaussian filter kernel was applied to the grayscale image in order to smooth and remove Gaussian noise from the images. An adaptive binary threshold was then applied to delineate the background from the PLA signal. The size of the neighborhood region that determines the threshold value for a given pixel was kept constant between control and experimental images. Finally, to determine the amount of newly synthesized protein, the processed image was passed through the connected components labeling software, which provides the total number of unique and continuous pixel groupings found in the image. One identified pixel represents a single nascent protein; therefore, the output corresponds to the total number of nascent proteins in the original image and can be used in conjunction with the total number of cells in the image to determine the average number of nascent proteins per cell.

## Statistical analysis

**Differential gene expression analysis and visualization.** Differential gene analysis was performed using DEBrowser, an R package, to detect significant changes in gene expression [68]. We conducted a paired analysis where expression levels between and differentiated cells were compared within each cell line in order to determine an overall significance level for GSCs and DGCs collectively. The voom function was first used to standardize the raw read counts and to apply precision weights for linear model analysis to account for the mean-

variance relationship observed in read counts [69]. The standardized reads were entered into DEBrowser. DESeq2 was used for differential gene expression analysis at an adjusted p-value cut-off of <0.05. Data visualization was performed using the DEbrowser heatmap and scatter-plot built in options. Additionally, Enrichr was used for pathway enrichment within the respective sets of upregulated and downregulated genes[70]. Lastly, certain boxplots were produced using the shiny app BoxPlotR[71].

**MeRIP-sequencing analysis.** To detect RNA methylation events, each exonic region was partitioned into contiguous 10-nucleotide segments. For each segment, the number of reads that mapped with a quality score of 20 or greater in each, the immunoprecipitated sample and the total RNA control sample, was determined using in-house scripts. The counts in each window were then analyzed, assuming a negative-binomial distribution, with the Fisher Exact Test in the R package, using as inputs the numbers of reads in the MeRIP and the control (non-immunoprecipitated) sample, and the corresponding total number of reads that mapped to the exonic regions of the genome for each sample. To make the procedure less sensitive to local variations, the mean number of counts for the entire exon of the control sample was used, instead of the counts in the 10-nucleotide window. The resulting p-values were adjusted using the Benjamini & Hochberg correction as implemented in the p.adjust function. To infer an RNA methylation signal, contiguous 10-nucleotide segments with a p-value smaller than an empirically determined threshold (see below) were joined together. Positive regions of at least 90 nucleotides in length were considered methylation events. A contiguous region longer than 200 nucleotides was assumed to have arisen from multiple methylation events near one another; such regions were split into 200-nucleotide segments, each representing a separate methylation event. To determine the p-value threshold for the selection of methylated regions, we looked at the number of events found different p-value levels, ranging from 1e-03 to 1e-40 and used a p-value that produced approximately 10,000 methylation events per sample: 1e-05 for four of the six samples and 1e-30 for the remaining two. Signal tracks for methylation events were plotted using the "*ChIPseeker*" package on R version 3.4.4 (R Foundation for Statistical Computing, Vienna, Austria).

**Quantification of qPCR, RIPs, Western Blots, PLA.** Our goal is to obtain results with greater than 95% confidence level. Assuming that data are normally distributed and that the standard deviation for measurements is no more than 3/4 of the mean, the t-test of mean was used to estimate the number of required observations. To determine significance among the means of three or more independent groups, we used one-way ANOVA. The homogeneity of variances was confirmed with Brown and Forsythe test, and the significance between specific groups was calculated with a post hoc Dunnett test. To determine significance among the means of two independent groups, we performed an unpaired two-tailed t test. To verify Gaussian distribution of data before applying the t test, we performed the D'Agostino and Pearson and Shapiro-Wilk normality tests.

## Supporting information

**S1 Text.** Supporting figures A-F. **Fig A: Characterization of CSCs and differentiated cells.** A) CSCs express stem cell specific transcripts (CD133, Sox2, Olig2), which they completely lose (CD133, Olig2) or downregulate (Sox2) after differentiation for 7 days by removal of EGF, bFGF and Heparin and addition of 10% serum. Moreover, following differentiation the glioma cells gain expression of GFAP, which was not expressed in CSCs. The graph presents representative RNA-seq data from one CSC line. The same analysis has been performed for all CSCs and differentiated glioma cells. B) Limiting dilution assay to determine the self-renewal ability of GSCs. The experiments were repeated six times and significance was calculated with a Chi-

square test (p<0.008). C) Orthotopic xenograft transplantation of CSCs in nude mice results in formation of invading glioblastomas, verifying the tumor-forming ability of the CSC lines. Image shows a HuNu positive glioblastoma 4 weeks after the transplantation of 150,000 CSCs. Hematoxylin was used as counterstain. Xenograft transplantations to examine tumor forming ability of CSCs are routinely performed for each newly isolated CSC line. D) Representative m6A dot blot following m6A RIP in GSCs and differentiated cells shows enrichment of m6A compared to input. E) Representative Ribo-seq profile of GSC. Ribo-seq reads distribution and median between stem and differentiated progenies showing that median values and read distribution are reproducible across experiments. **Fig B: Methylation and Transcriptome Profile.** A) Distribution of genome wide m6A peaks in GSCs and differentiated progeny divided in 5'UTR, CDS and 3'UTR peak regions. B) C) D) Mean m6A change vs. TE FC scatterplot and transcripts dichotomized into m6A loss (n = 1382) and gain (n = 1455) groups based on mean m6A change during differentiation. Transcripts TE distributions shown by mean m6A loss and gain in individual GSCs and corresponding DGCs. **Fig C:** A) Log2 TE comparison of GSCs and DGCs on transcripts with change in TE rank $\geq$ 60th percentile, demarking region of median m6A change (n = 11,179) (Wilcoxon test). B) M6A distribution in GSCs and DGCs on transcripts with change in TE rank $\geq$ 60th percentile. C) TE log2 FC in transcripts grouped according to percent m6A loss (A: <50% loss, B: 50–75% loss, C: >75% loss) (m6A loss obtained by subtracting initial total peak (GSC) from final amount (DGC). Captures mean/general changes in TE using log2 FC between GSCs and DGCs. D) log2 TE of transcripts with $\geq$ 2 peaks loss and change in TE rank $\geq$ 70th percentile (n = 568, n = 410, n = 489; GSC1/DGC1,2,3 respectively; Wilcoxon test). E) Scatterplot of differential expression between GSCs and DGCs, 128 common transcripts in green. **Fig D:** A) Upregulated genes in DGCs compared to GSCs from RNA-seq expression analysis (red), exhibit RNA pol II reads that are statistically higher in DGCs than those in GSCs. Similarly, for the downregulated set of genes in RNA-seq (blue), the RNA pol II reads are statistically lower than those in GSCs (p<0.05). B) Heatmaps of RNA Pol II ChIP-seq signal at genes found to be up- (red) or downregulated (blue) using RNA-seq in GSCs versus DGCs. Genes are ranked from most upregulated to least (at left) and least downregulated to most (at right), demonstrating that RNA Pol II ChIP-seq signal scales with results from RNA-seq. C) Example regions of RNA Pol II ChIP-seq peaks on GSCs and DGCs. D) We calculated the change in TE percentile between the 128 common transcripts and other transcripts with top 30% increased TE percentile. The top transcripts with the top 30% increased TE during differentiation were collected and grouped into 128 common transcripts and others (transcripts in the top 30% but that did not follow the m6a/TE trend across all GSCs). Wilcoxon test was performed on the change in TE percentile of the 128 common transcripts versus other top 30% non-common transcripts across all GSCs. The 128 common transcripts experience the greatest increase in TE amongst the top 30% most efficiently translated transcripts. E) We determined the fraction of transcripts with m6a loss and increase in TE whose miRNA binding sequence overlaps a RRACH motif. All transcripts with m6a loss were collected per patient. A group of transcripts with the top 30% TE percentile increase that have undergone significant peak loss (equal to or greater than 2 peak loss) were obtained (GSC1: 568 /3059; GSC2: 410/2115; GSC3: 489/2360). Of these transcripts, the majority, between 97% to 98%, was found to have a RRACH motif sequence and from those with a RRACH motif, between 24% and 35% had a RRACH motif overlapping a miRNA binding sequence. (GSC1: 134/553; GSC2: 141/404; GSC3: 119/481). Key findings: 24% to 35% of the transcripts that experience m6a loss and increase in TE during GSCs to DGCs transition have miRNA binding sequence overlapping the RRACH motif. **Fig E:** A) Expression level of transcripts constituting the cellular m6A machinery do not change during GSC differentiation. B) Protein levels of the m6A erasers FTO, Alkbh5 and writers Mettl3, Mettl14 do not change

during GSC differentiation. C) A bipartite network depicting miRNA with predicted transcript targets. All miRNAs predicted to bind within m6A enriched RRACH motifs. D) CLIP3 Survival and expression data across three TCGA platforms. E) Expression of RRACH-binding miR-143 and miR-129 induces significant increase in association of FTO with the corresponding targeted mRNAs. **Fig F:** A) Transfection of DGCs with miR-145 antagomir inhibits expression of miR-145 as determined by qRT-PCR. B) YTHDF2 transcript expression in GSCs and DGCs as determined by RNA-seq using three patient-derived cell lines. C) Inhibition of miR-145 expression with miR145 antagomir does not affect transcript expression levels of CLIP3 as demonstrated by qRT-PCR. D) Representative images of AHA Pulse-chase to detect nascent translation of YTHDF2 after transfection of DGCs with a non-targeting antagomir (left panel) or a miR145-specific antagomir. Inhibition of miR-145 induces nascent translation of YTHDF2. E) Quantification of number of PLA dots per cell shows that inhibition of miR-145 expression using a miR-145 antagomir results in significant increase of nascent translation of YTHDF2. Significance was calculated from at least 200 cells per condition using a Student's t-test (**p<0.0005, n = 3 biological replicates). F) Antibody only control of PLA presented on Fig 5B shows lack of non-specific signal.
(DOCX)

## Author Contributions

**Conceptualization:** Nikos Tapinos.

**Formal analysis:** David Karambizi, J. Eduardo Fajardo, Oliver Y. Tang, Jia-Shu Chen, Andras Fiser, Nikos Tapinos.

**Funding acquisition:** Steven A. Toms, Nikos Tapinos.

**Investigation:** John P. Zepecki, David Karambizi, Kristin M. Snyder, Charlotte Guetta-Terrier, Atom Sarkar, Nikos Tapinos.

**Methodology:** John P. Zepecki, Nikos Tapinos.

**Supervision:** Nikos Tapinos.

**Validation:** Nikos Tapinos.

**Visualization:** David Karambizi.

**Writing – original draft:** Nikos Tapinos.

**Writing – review & editing:** John P. Zepecki, David Karambizi, Steven A. Toms, Nikos Tapinos.

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
