## [Decision Letter · Decision Letter 0]

7 Oct 2020

Dear Dr Tapinos,

Thank you very much for submitting your Research Article entitled 'miRNA-mediated loss of m6A increases nascent translation in glioblastoma' to PLOS Genetics. Your manuscript was fully evaluated at the editorial level and by independent peer reviewers. The reviewers appreciated the attention to an important topic but identified some aspects of the manuscript that should be improved.

We therefore ask you to modify the manuscript according to the review recommendations before we can consider your manuscript for acceptance. Your revisions should address the specific points made by each reviewer.

We hope to receive your revised manuscript within the next 60 days. If you anticipate any delay in its return, we would ask you to let us know the expected resubmission date by email to plosgenetics@plos.org.

[LINK]

Yours sincerely,

Alexey Ruzov

Guest Editor

PLOS Genetics

Wendy Bickmore

Section Editor: Epigenetics

PLOS Genetics

Reviewer's Responses to Questions

**Comments to the Authors:**

Reviewer #1: Zepecki et al. manuscript compares gene expression between glioma stem cells (GSCs) and differentiated glioma cells (DGCs) together with the analysis of RNA m6A modification which is known to affect transcript stability and expression. They find a subset of transcripts that have reduced m6A levels and increased translation between GSCs and DGCs. Further analysis identifies that these transcripts contain sequences that match certain miRNAs and changing the expression of miRNAs affects the methylation of target RNAs. Zepecki et al., proposes that miRNA targeting of certain transcripts can lead to their demethylation and increased translation without affecting mRNA abundance. This is an important finding and valuable addition to the mechanisms that regulate gene expression via miRNAs and RNA modifications. Overall, the manuscript presents substantial amount of data to support the direction of their claims. Except a few overstatements and lack of controls presented, manuscript would be an important addition to RNA field. I find the proposed mechanism highly compelling.

I would recommend that if the authors can provide the essential controls requested below or an explanation clarifying why they are not necessary, the manuscript should be considered for publication at PLoS Genetics. Please note that I wasn’t able to check if RNA sequencing data is available on a database.

Specific comments are below;

1- At the heart of the manuscript is the MeRIP-Seq experiment conducted to identify m6A methylated transcripts. Multiple figures do not contain any validation of the MeRIP-Seq approach. In particular, can the authors show that they are effectively pulling down m6A methylated transcripts using ideally mass spectrometry to show enrichment of m6A in pull-down vs input OR at least a dot blot approach to indicate to some level their IP is working. Can the authors document if the efficiency of m6A IP compatible between experiments in different figures (figure In Supp. Fig 2C, how does the input look like?

2- Similarly, can the authors show that their ribosome profiling experiment is working as expected by showing actually the profile between experiments. As it stands the manuscript mentions both MeRIP-Seq and Ribosome profiling experiments were conducted without actually showing any validation that these methods work in the hands of the authors to a degree that is comparable with other studies.

3- What is the relation between m6A change and transcript abundance between GSCs and DGCs?

4- “Although median increase in TE was noted with both m6A loss and gain, there is a relatively more substantial increase in median TE in all DGCs in the subset of transcripts with mean m6A loss. In comparison, the transcripts with mean increase in m6A show a blunted non- generalized, increase in median TE (Supplementary Figure 2D). Taken together, these findings suggest a stronger link between m6A loss and TE increase during GSC differentiation.”

From the above statement it is not clear (1) if there is a statistically significant difference in TE among genes that either loose or gain at least one m6A peak and (2) what authors mean by “blunted non-generalised increase” in contrast to “relatively substantial increase”. I would expect to see some statistical tests to test hypothesis here and more clear language.

5- In supplemental figure 3, authors show that for the top 40% genes, there is a reduction in m6A and increase in TE between GSCs and DGCs. This is rather confusing, are the authors taking top 40% genes that show TE increase and plot that they indeed have TE increase? What about the rest of the genes? What is the profile for m6A status?

6- If a gene is showing increased translation efficiency, how likely it is that the said gene looses its m6A more than 75% as opposed to showing no change or increase in m6A?

7- “In addition to the 70th percentile cut-off, we imposed a methylation cut-off of 2 peak loss in order to capture all transcripts significantly fitting the m6A loss and increase in TE trend.”

Previously authors used a 60th percentile and at least 1 m6A loss as a criteria. Why the need for change? This is not sufficiently explained.

8- Throughout the manuscripts authors suggest that the m6A is lost on transcripts. It is important the authors clarify if they mean these transcripts are demetyhlated or they are not methylated when new transcripts are made. I suppose this cannot be distinguished with current data.

9- No data is presented to show that RNA pol II ChIP experiments are working effectively across samples. Can the authors at least provide example regions with input and RNA pol II read peaks?

10- In Figure 3, authors should also analyse all m6a regions and compare to 128 transcripts that follow a desired trend. This would provide evidence if these transcripts are unique and regulated differently or similar miRNA binding sites can be found in all m6A transcripts which would indicate additional mechanisms for the 128 transcripts.

11- Figure 4A should include anti-FTO western blot to show that FTO is expressed, upon IP it is depleted from input and enriched in IP fraction to show that the IP is working effectively. In addition, is the FTO - AGO1 interaction RNA dependent? To test this authors should do the IP in the presence and absence of RNAse digestion.

12- For Figures 4D, and Supp. Fig 5E what is the proof that authors are not immunoprecipitating more FTO after miRNA over expression?

13- In Figure 5B and E, authors should present the single antibody only controls to clearly present the background level signal.

14- Figure 5E, can the authors show that the mir-145 antagomir effect is specific to CLIP3 translation and not generally affecting all translation in the cells?

15- Overall, the methods section lacks sufficient detail in some parts;

- m6A RNA IP section; how is it possible to repeat these experiments with the information given here? How much antibody? What were the controls? How much RNA?

- where is the method for polysome fractionations?

- western blots; what are the antibody concentrations? How much protein loaded?

- etc.

Reviewer #2: Please check attachement.

**Have all data underlying the figures and results presented in the manuscript been provided?**

Reviewer #1: **No: **I couldn't access to the GEO database to check if all data is available. The scripts are not made available.

Reviewer #2: Yes

PLOS authors have the option to publish the peer review history of their article (what does this mean?). If published, this will include your full peer review and any attached files.

Reviewer #1: No

Reviewer #2: No

---

## [Decision Letter · Decision Letter 1]

15 Feb 2021

Dear Dr Tapinos,

We are pleased to inform you that your manuscript entitled "miRNA-mediated loss of m6A increases nascent translation in glioblastoma" has been editorially accepted for publication in PLOS Genetics. Congratulations!

Yours sincerely,

Wendy A. Bickmore

Section Editor: Epigenetics

PLOS Genetics

Comments from the reviewers (if applicable):

Reviewer's Responses to Questions

**Comments to the Authors:**

Reviewer #1: Zepecki et al., has substantially revised the original manuscript and addressed all questions raised during the initial review process. In addition, Zepecki et al., has carried out additional experiments and analysis which further strengthen the message of the manuscript. They have updated multiple figures with addition of new data, and also present many more control experiments. Overall, the manuscript is much improved and in my opinion should be accepted for publication in PLoS Genetics.

**Have all data underlying the figures and results presented in the manuscript been provided?**

Reviewer #1: Yes

PLOS authors have the option to publish the peer review history of their article (what does this mean?). If published, this will include your full peer review and any attached files.

Reviewer #1: No

**Data Deposition**

http://datadryad.org/submit?journalID=pgenetics&manu=PGENETICS-D-20-01310R1

**Press Queries**

---

## [Editor Report · Acceptance letter]

26 Feb 2021

PGENETICS-D-20-01310R1 

miRNA-mediated loss of m6A increases nascent translation in glioblastoma 

Dear Dr Tapinos, 

We are pleased to inform you that your manuscript entitled "miRNA-mediated loss of m6A increases nascent translation in glioblastoma" has been formally accepted for publication in PLOS Genetics! Your manuscript is now with our production department and you will be notified of the publication date in due course.

With kind regards,

Alice Ellingham

PLOS Genetics

On behalf of:
